# Immunological Control of Herpes Simplex Virus Type 1 Infection: A Non-Thermal Plasma-Based Approach

**DOI:** 10.3390/v17050600

**Published:** 2025-04-23

**Authors:** Julia Sutter, Jennifer L. Hope, Brian Wigdahl, Vandana Miller, Fred C. Krebs

**Affiliations:** 1Center for Molecular Virology and Gene Therapy, Institute for Molecular Medicine and Infectious Disease, and Department of Microbiology and Immunology, Drexel University College of Medicine, Philadelphia, PA 19102, USA; js4932@drexel.edu (J.S.); jlh434@drexel.edu (J.L.H.); bw45@drexel.edu (B.W.); vam54@drexel.edu (V.M.); 2Immune Cell Regulation and Targeting Program, Sidney Kimmel Comprehensive Cancer Center Consortium at Jefferson Health, Philadelphia, PA 19107, USA

**Keywords:** non-thermal plasma, cold atmospheric plasma, gas plasma, immunotherapy, antiviral, CD8^+^ T cell, latent infection, reactivation

## Abstract

Herpes simplex virus type 1 (HSV-1) causes a lifelong infection due to latency established in the trigeminal ganglia, which is the source of recurrent outbreaks of cold sores. The lifelong persistence of HSV-1 is further facilitated by the lack of cure strategies, unsuccessful vaccine development, and the inability of the host immune system to clear HSV-1. Despite the inefficiencies of the immune system, the course of HSV-1 infection remains under strict immunological control. Specifically, HSV-1 is controlled by a CD8^+^ T cell response that is cytotoxic to HSV-1-infected cells, restricts acute infection, and uses noncytolytic mechanisms to suppress reactivation in the TG. When this CD8^+^ T cell response is disrupted, reactivation of latent HSV-1 occurs. With antiviral therapies unable to cure HSV-1 and prophylactic vaccine strategies failing to stimulate a protective response, we propose non-thermal plasma (NTP) as a potential therapy effective against recurrent HSV-1 infection. We have demonstrated that NTP, when applied directly to HSV-1-infected cells, has antiviral effects and stimulates cellular stress and immunomodulatory responses. We further propose that the direct effects of NTP will lead to long-lasting indirect effects such as reduced viral seeding into the TG and enhanced HSV-1-specific CD8^+^ T cell responses that exert greater immune control over HSV-1 infection.

## 1. Introduction

Herpes simplex virus type 1 (HSV-1) is a highly contagious lifelong pathogen that poses a large global health burden. Currently, an estimated 3.8 billion people under the age of 50 are infected with HSV-1, impacting ~70% of the world’s population [1]. While HSV-1 infection is primarily asymptomatic, infected individuals may experience periodic outbreaks of symptomatic infection characterized by the development of small, painful cold sores that appear around the mouth, eyes, and genitalia. These periodic outbreaks of symptomatic infection are caused by the reactivation of latent reservoirs that HSV-1 establishes in the trigeminal ganglia (TG) or the dorsal root ganglia (depending on the site of transmission), allowing lifelong viral persistence [2,3]. Access to the nervous system can cause HSV-1 to disseminate into central nervous system (CNS) tissues to cause complications like encephalitis [4]. Additionally, emerging evidence has demonstrated a correlation between chronic HSV-1 infection and the onset of Alzheimer’s disease in elderly populations [5,6].

Lifelong persistence of HSV-1 is further aided by the lack of curative treatments available for infected individuals. Current standard of care (SOC) therapies, while effective in enhancing the resolution of symptomatic infection, are unable to prevent the establishment of latent reservoirs and subsequent reactivation events [7]. These limitations can be troubling for immunocompromised individuals, who are more susceptible to recurrent outbreaks and to HSV-1-associated disease due to their weakened immunological control over infection.

The course of HSV-1 infection is heavily controlled by the host’s adaptive immune system. In this review, the multiple mechanisms of how the host’s immune system limits symptomatic infection in the periphery and suppresses reactivation of asymptomatic latent infection are discussed. Additionally, we discuss the limitations of the host immune system that prevent clearance of HSV-1 and lead to recurrent outbreaks over time in individuals. Furthermore, we explore a unique form of therapy administered at the local lesion that has antiviral effects on HSV-1 infection and immunomodulatory effects that will result in the more effective targeting of HSV-1-infected cells at both acute and latent infection sites.

## 2. HSV-1 as the Etiological Agent of Herpes Labialis

Oral herpes, or herpes labialis, is a common outcome of HSV-1 infection and accounts for the majority of HSV-1 infections. Herpes labialis is characterized by the presence of cold sores that appear around and inside the mouth. Transmission of HSV-1 infection via an oral route occurs following direct or indirect contact with an oral cold sore or saliva from an infected individual. At the site of transmission, which is typically around the lips, HSV-1 replicates and propagates in mucosal epithelial cells, leading to the formation of cold sores [2,3]. Simultaneously, HSV-1 enters nearby sensory neurons and is transported to the TG, where it establishes a latent infection. As acute infection in the periphery resolves, patients become asymptomatic for infection but retain HSV-1 inside latently infected sensory neurons, which provide a reservoir for future reactivation (Figure 1). Reactivation is often triggered by external and internal stress stimuli that induce the release of HSV-1 from the TG and recurrence of acute infection at the initial transmission site [8].

### 2.1. Acute Epithelial Infection in Herpes Labialis

The primary cells that support HSV-1 replication and propagation are mucosal epithelial cells and epidermal keratinocytes at the site of initial transmission [9]. At the site of acute infection, the envelope glycoproteins initiate attachment to cellular proteoglycans, such as heparan sulfate, on mucosal epithelial cells at the site of transmission. Other cellular receptors, such as nectin-1 and herpesvirus entry mediator (HVEM), can also facilitate HSV-1 entry [10,11]. This attachment process is often targeted by the host’s adaptive immune system, as the viral glycoproteins involved in HSV-1 entry, gB and gD, are often the immunodominant antigens for HSV-1-specific T cells [12,13]. Following attachment, a pH-mediated endocytosis viral entry process occurs that releases the encapsulated viral genome into the infected cell [14]. Tegument proteins, included in the HSV-1 virion, are also released into the cell to mediate stages of the HSV-1 replication cycle [15,16] and affect host cell immune signaling cascades as a viral evasion tactic [17,18,19]. Productive replication in mucosal epithelial cells and epidermal keratinocytes results in the release of virions that infect nearby cells and propagate the infection.

During oral infection, HSV-1 infection is not limited to a single cell type. HSV-1 has a broad tropism, having the ability to infect multiple cell types within the oral mucosal tissue. These cell types include responding immune cells, like dendritic cells (DCs), which play critical roles in immune surveillance and activating the adaptive immune response. However, the infection of immune cells often leads to an incomplete replication cycle (an abortive infection), indicating a possible evasion mechanism by HSV-1 to downregulate the function of innate immune cells during acute infection [20,21].

Clinical symptoms of acute epithelial infection present as orofacial cold sores. The initial symptoms of HSV-1 infection can appear up to six hours post-exposure or after a reactivation event as a result of rapid propagation by the virus in epithelial cells. In this initial phase, referred to as prodrome, infected individuals may experience a tingling sensation, burning, and/or itching at the acute infection site. Eventually, these initial symptoms lead to the development of ulcerative lesions with erythema [2,3]. With direct contact, these lesions can facilitate the transmission of the virus.

### 2.2. Establishment and Maintenance of Latent Virus Reservoirs in Neurons

The establishment of latent infection occurs simultaneously with acute infection. When the virus becomes entirely latent, clinical symptoms associated with acute infection disappear and the patient becomes asymptomatic for the infection. During acute infection, HSV-1 enters nearby sensory neurons and travels into the TG to establish a persistent latent infection. Similar to acute infection, both attachment and entry into sensory neurons are mediated by the viral glycoproteins that bind to cellular receptors, heparan sulfate, or nectin-1, mediating pH-independent fusion into the cell [22,23]. The encapsulated viral genome then undergoes retrograde transport toward the cell body using the neuron’s cytoskeletal network [22,24]. At the cell body, HSV-1 expresses a latency-associated transcript (LAT) to establish a latent infection, characterized by the absence of HSV-1 replication and virus production [25,26,27,28,29]. While LAT was originally thought to be expressed under a neuron-specific promoter, there is evidence of LAT expression in non-neuronal cells, complicating our understanding of latency [30].

LAT is the hallmark of latent infection and is expressed from a non-integrated episome associated with cellular histones, allowing transcriptional control over the viral genome. This transcriptional control over replication-associated genes is partly mediated by a LAT-encoded microRNA that directly suppresses viral gene expression [31,32]. LAT also plays direct roles in regulating histone methylation on replication-associated genes. In comparing LAT^+^ and LAT-null HSV-1 strains, LAT^+^ strains were found to increase dimethyl lysine 9 on histone 3 of the viral genome, promoting heterochromatin formation over the genes associated with replication during acute infection, silencing their expression [33,34]. LAT also promotes the expression of cellular genes that prevent apoptosis in the infected neuron, ensuring HSV-1 survival. One study suggests that LAT downregulates the JAK-STAT pathway in neurons and interferes with the expression of type I IFNs as a mechanism to inhibit apoptosis [35]. Altogether, the function of LAT aids in the maintenance of latent infection, evasion of the host immune response, and HSV-1 persistence in the host.

### 2.3. Reactivation from Latent Infection

Patients living with herpes labialis are subjected to frequent outbreaks of symptomatic infection as a direct result of reactivation. HSV-1 infection is reactivated by various pleotropic stressors such as exposure to ultraviolet light, fever, menstruation, psychological stress, superinfection, and skin trauma [8,36,37,38,39,40,41]. Exposure to these stressors prompts the anterograde transport of HSV-1 from sensory neurons to resume infection of mucosal epithelial cells at the initial site of transmission and neuron innervation, leading to the re-appearance of cold sores [42,43,44]. Symptoms associated with recurrent oral herpes infections are generally milder relative to the symptoms that arise during the primary infection.

Reactivation of HSV-1 from latently infected neurons in the TG is thought to be mediated by a balance of factors. The most accepted trigger of reactivation from latency is stress stimuli [8]. One study implicates stress stimuli in the activation of a neuron-specific Jun-N-terminal kinase (JNK) pathway that mediates the histone phospho/methyl switch responsible for control over the transcriptional activation of replication-associated genes suppressed during latency [45]. Other studies suggest that stress stimuli increase corticosteroids that bind and activate glucocorticoid receptors (GRs). Upon the activation of GRs, the receptor translocates into the nucleus to bind the glucocorticoid response element to prompt chromatin remodeling and activate the transcription of replication-associated genes [46,47]. Evidence of LAT disruption was also observed when ganglionic cultures were treated with proapoptotic drugs, inducing the expression of many genes associated with acute infection [48]. While many mechanisms may be involved, the disruption of LAT’s ability to maintain latency is a common feature in reactivation

## 3. Host Responses and Immune Control of Herpes Labialis

The pathogenesis of HSV-1 is tightly controlled by the host immune system. This immune control occurs at two locations: the site of acute infection and the site of latent infection. At the acute site of infection, initial exposure to HSV-1 leads to two types of immune response. Innate immune responses are immediate upon exposure to HSV-1, playing a critical role in detecting HSV-1 antigens and activating the adaptive immune response. Adaptive immune responses, which are more specific to HSV-1, are delayed and become active during the later stages of acute infection. The adaptive immune response is also dominant during re-exposure to HSV-1 when the virus reactivates and resumes acute infection at the oral mucosa. Primarily, this adaptative immune response involves HSV-1-specific CD8^+^ T cells that selectively target HSV-1-infected cells to control the spread of HSV-1 and shorten the symptomatic window of acute infection. Adaptive immune responses are also active in the TG during latent infection. While CD8^+^ T cells are unable to clear latently infected sensory neurons, their immune surveillance suppresses spontaneous reactivation of latent HSV-1 and prevents dissemination of HSV-1 into distant tissues [49]. This immunological control is maintained until HSV-1 reactivates and re-enters into the periphery to resume symptomatic infection. Additionally, chronic HSV-1 infection can cause inflammation in the nervous system and lead to HSV-1-associated disease in some cases. Although immunocompromised hosts are more susceptible to HSV-1-associated complications, all infected individuals can be at risk [4,5] (Figure 2).

### 3.1. Innate Immune Responses to HSV-1 Transmission and Acute Infection

The oral mucosa consists of mucous membranes that line the lip and the mouth, serving as the first line of defense against pathogens like HSV-1. The mucosal membrane is composed of stratified epithelium with a keratinized layer of squamous epithelial cells over a layer of connective tissues called the lamina propria [50,51]. Epithelial cells and keratinocytes within the oral mucosa, which serve as the target cells for HSV-1, contain many pathogen recognition receptors (PRRs) and DNA sensors [52,53]. These sensors are capable of detecting antigens that can trigger immune signaling pathways, leading to the expression of pro-inflammatory cytokines, chemokines [53,54], and antimicrobial peptides [55]. The mucosal membranes also contain various immune cells, like DCs and Langerhans cells (LCs), that can serve as antigen-presenting cells (APC) to activate antigen-specific T cell responses in the draining lymph nodes [56,57,58].

#### 3.1.1. HSV-1 Is Detected by Cellular Sensors During Acute Infection

Immunological control over HSV-1 infection is activated during acute infection through antigen sensing by PRRs. PRRs are receptors that are located on the cell membrane or in intracellular compartments of infected cells. They play major roles in immune surveillance by binding pathogen-associated molecular patterns (PAMPs), conserved antigenic motifs, or damage-associated molecular patterns (DAMPs), molecules released by cells undergoing stress [59,60,61] and HSV-1 infection [62]. Upon binding to PRRs, signaling cascades are activated that result in the production of pro-inflammatory cytokines and interferons (IFNs) by cells [61,63]. Cytokines and IFNs are critical for inducing inflammatory responses that recruit innate immune cells to the infection site. Meanwhile, IFNs can restrict viral replication in keratinocytes [64,65] and inhibit the release of HSV-1 from infected cells [66]. IFNs can also stimulate the antiviral activity of many innate immune cells, such as natural killer (NK) cells, macrophages, and T cells [63].

Toll-like receptors (TLRs) play particularly critical roles in activating the innate immune response against HSV-1. TLRs are a class of PRRs found extracellularly on the plasma membrane, and intracellularly in endosomal compartments and the cytoplasm [67,68]. Specifically, TLR2 is a common sensor for HSV-1 that can detect intact virions by binding to envelope glycoproteins, gH/gL [69]. TLR2 is located on both the cell membrane and within the cytoplasm of the infected cell. After antigen sensing, TLR2 mediates the production of pro-inflammatory cytokines and chemokines by activating the nuclear factor-κB (NF-κB) pathway through myeloid differentiation factor 88 (MyD88)/ tumor necrosis factor receptor associated factor 6 (TRAF6) signaling [60,61,70]. Some studies suggest that TLR2 requires dual recognition of HSV-1 antigens with endosomal TLR9 [71,72]. Like TLR2, TLR9 promotes the activation of the NF-κB pathway [73]. However, TLR9-knockout studies suggest that its activation is dispensable for the control of HSV- infection [74]. With TLR2, TLR9 works synergistically to promote the expression of pro-inflammatory cytokines and IFN-stimulated genes (ISG) [75].

Lastly, endosomal TLR3 can also activate the antiviral immune response against HSV-1 by sensing pathogenic double stranded RNA [76]. Activation of TLR3 leads to a type 1 IFN response, which can be initiated through Myd88-independent signaling pathways [77] or by Myd88/TRAF6-dependent activation of NF-κB and IRF3 [78]. While the mechanisms of HSV-1 dsRNA detection are not fully understood, the deficiencies of TLR3 are associated with greater HSV-1 infection severity [79] as well as higher incidences of HSV-1-induced encephalitis [80].

In addition to TLRs, other DNA/RNA sensors have been implicated in the sensing of HSV-1 infection. Cyclic GMP-AMP synthase (cGas) is a cytoplasmic DNA sensor found in keratinocytes that can bind HSV-1 DNA and trigger the production of type 1 IFN through a stimulator of interferon genes (STING)-dependent mechanism via TANK-binding kinase 1 (TBK1) signaling [81,82]. While cGas serves as a source of type I IFNs during acute infection, deficiencies are associated with a higher encephalitis risk in infected mice [83]. IFN-γ-inducible protein (IFI16) is another viral DNA sensor that detects HSV-1 in the nucleus of infected cells. Like cGas, IFI16 produces type I IFN through the STING/TBK1-dependent signaling pathway [52]. Additionally, retinoic acid-inducible gene-1 (RIG-1) receptors, such as melanoma differentiation-associated protein 5 (MDA5) and RIG-1, have been shown to sense HSV-1 dsRNA to produce both type I and type III IFNs via NF-κB and IRF3/7 activation [61,84]. Although it is unclear how HSV-1 dsRNA accumulates in the cytoplasm for RIG-1 sensing during the replication of this DNA virus, some evidence of RIG-1 binding to cytosolic HSV-1 DNA through RNA pol III has been reported [61,85].

#### 3.1.2. Innate Immune Cells Mediate Antiviral Immune Responses Against HSV-1 and Promote Adaptive Immunity

The host cell receptors and sensors involved in the detection of HSV-1 during acute infection all stimulate pro-inflammatory responses and IFN production. Inflammation at the site of infection can promote the recruitment of innate immune cells for innate killing, antigen uptake, and downstream activation of the adaptive immune response.

Myeloid cells are major contributors to the innate defense against invading viral pathogens in tissues. Neutrophils, which are the first white blood cell type recruited to the infection site, add to the inflammatory response to HSV-1 and can also transport viral antigens to the draining lymph nodes for activation of T cells [86]. The presence of neutrophils has been reported at sites of acute as well as latent infection in HSV-1 cutaneous and cornea infection models [87,88]. While neutrophils stimulate enhanced T cell activity in influenza virus infection models, they are suggested to be dispensable for HSV-1 infection and do not contribute significantly to the activation of the adaptive immune response [89]. Other myeloid cells, such as M1 macrophages, were found to protect mice against HSV-1 corneal infection [90]. Macrophages reside in all tissues and likely mediate their antiviral responses by internalizing viral particles and restricting infection in a type I IFN-dependent manner [91,92].

NK cells are another white blood cell type that are early responders to HSV-1 acute infection [93]. Unlike myeloid cells, NK cells directly lyse and trigger cell death in HSV-1-infected cells. On their cell surface, NK cells possess inhibitory and activating receptors that bind to MHC class I molecules on target cells [94]. As a viral evasion tactic, the HSV-1 ICP47 protein downregulates MHC class I molecules on infected cells to limit antigen presentation [95]. This downregulation is detected by NK cells, triggering the secretion of IFNγ and cytolytic granules, perforin, and granzyme B, which form pores in the target cell membrane for the induction of apoptosis [96,97]. However, the contribution of NK cells to mediating the antiviral response against HSV-1 is debatable. Many early studies correlated reduced NK cell activity with severe HSV-1 infection and higher mortality rates [98,99,100]. Their roles in enhancing T cell responses during HSV-1 infection have also been reported [101]. In contrast, some studies showed NK cell activity to not contribute to improved clinical outcomes [102].

DCs play key roles as innate immune cells during HSV-1 infection. DCs can be classified into two major subsets: conventional (cDC) and plasmacytoid (pDC). pDCs are recruited early to the infection site and mediate antiviral activity through the secretion of type I IFN, particularly IFNα, following antigen sensing by TLRs [103,104,105]. cDCs, in contrast, are migratory, shuttling antigens from the infection site to the draining lymph nodes for presentation to T lymphocytes. Unlike other APCs, cDCs can present antigens on both MHC class I and class II molecules to activate CD8^+^ and CD4^+^ T cells, respectively [106]. The depletion of pDCs in transgenic mouse models was associated with severe HSV-1 infection. This effect was likely due to less innate immune cell recruitment and activation, which was found by downregulating the expression of IFNγ and IFNα [107,108,109,110]. Additionally, a model for HSV-1-induced acute retinal necrosis indicated an association between the depletion of DCs and lower CD8^+^ T cell activation, confirming the role of DCs in the later activation of the adaptive immune response [111].

### 3.2. Adaptive Immune Responses to Productive Infection

Cell-mediated immunity is the dominant adaptive immune response that controls HSV-1 infection. T cells become activated following the recognition of antigens presented on MHC molecules. Following maturation in the thymus, T cells migrate to secondary lymphoid organs, where they wait for activation signals. These signals include the recognition of antigens by T cell receptors (TCRs) on MHC molecules by APCs, co-stimulatory signal, and, in some cases, activating cytokines. The T cell response to HSV-1 infection is primarily mediated by CD8^+^ cytotoxic T cells, which recognize endogenous antigens on MHC class I molecules, and CD4^+^ helper T cells, which are activated by the exogenous antigens presented on MHC class II molecules. Once activated, T cells migrate to the site of infection to elicit specific cellular immune responses to viral infection [112,113].

Much of what we know about the role of the adaptive immune response on HSV-1 infection has come from mouse models of infection and pathogenesis. In a flank model of HSV-1 infection, C57/B6 mice demonstrated clearance of HSV-1 acute infection as early as 5 days PI, with gB-specific CD8^+^ T cells being found at infection site tissues. Similarly, gB-specific CD8^+^ T cells were capable of resolving HSV-1 acute infection in RAG^-/-^ mice [114]. Meanwhile, CD4^+^ T cells were found to peak around 6–8 days post-infection in the lymph nodes of mice using a genital infection of HSV-1 [115]. While the CD8^+^ T cell response is speculated to be dominant in clearing HSV-1, both CD8^+^ and CD4^+^ T cells are capable of clearing HSV-1 from the acute infection site.

#### 3.2.1. CD8^+^ T Cells

CD8^+^ T cells are effective in eliminating intracellular pathogens, implicating their importance in targeting virally infected cells. The activation of CD8^+^ T cells is antigen-specific, with the immunodominant peptide being gB for HSV-1 infections, as observed in C57/B6 mouse infection models [12,116]. These antigens are loaded onto MHC class I molecules by APCs, with DCs being the most notable for HSV-1 infection [117,118]. These antigens are then recognized by TCRs on naïve CD8^+^ T cells. The priming of CD8^+^ T cells is further achieved through co-stimulatory signals and by pro-inflammatory cytokines, like IL-12 and type I IFNs. These signals prompt the differentiation and proliferation of cytotoxic CD8^+^ T cells, allowing them to infiltrate into non-lymphoid tissues to respond to infection [119,120].

The cytolytic activity of CD8^+^ T cells is mediated by the release of two proteins, perforin and granzyme B [121]. Perforin mediates pore formation in the membrane of the HSV-1-infected cell, disrupting cell membrane integrity. In granzyme B-deficient CD8^+^ T cells, perforin alone is capable of inducing necrosis in virus-infected cells [122]. However, and in most cases, pore formation by perforin allows the entry of granzyme B into the infected cell to either directly or indirectly activate cell death pathways [123,124,125].

CD8^+^ T cells also produce IFNγ within hours of infection as part of their cytolytic mechanism [126]. IFNγ is critical in promoting the differentiation of HSV-1-spexific CD8^+^ T cells toward a memory phenotype [127]. CD8^+^ T cells also use IFNγ to elicit their cytotoxic killing of HSV-1-infected cells. In an environment deficient in IFNγ, CD8^+^ T cells were found to have more migratory potential and enhanced cytolytic function in both in vitro and in vivo HSV-1 infection models [128].

#### 3.2.2. CD4^+^ T Cells

CD4^+^ T cells also play a key role in viral infections by producing cytokines, promoting memory differentiation, and mediating the activation of adaptive immune cells, including CD8^+^ T cells and B cells. Like CD8^+^ T cell activation, CD4^+^ T cell activation is antigen specific, requiring TCR engagement with antigens presented on MHC class II molecules by APCs and a co-stimulatory signal [129,130]. In the context of HSV-1 infection, the immunodominant peptides for HSV-1-specific CD4^+^ T cells are the glycoproteins gD and gB [13,131]. Activated CD4^+^ T cells have been shown to exert immunological control over many herpesvirus infections by recognizing viral glycoproteins, inhibiting latency establishment, and participating in the direct killing of virus-infected cells [132,133,134].

Although the roles of T cells in herpes labialis have not been thoroughly examined, studies of other types of HSV-1 infection have revealed the protective effects of CD4^+^ T cells. For example, CD4^+^ T cells have demonstrated protective roles in ocular HSV-1 disease models. CD4^+^ T cell presence is strongly correlated with the enhanced clearance of acute HSV-1 infection and the reduced development of herpes simplex keratitis [135]. Intravaginal infection models for HSV-1 have also identified a role for CD4^+^ T cells in the clearance of HSV-1 in both genital and neural tissues [115]. Even though the role of CD4^+^ T cells has not been extensively studied in oral HSV-1 infection models, their presence was noted in the TG of HSV-1-infected mice at 15 days post-infection using the lip scarification model [136].

Upon activation, CD4^+^ T cells respond to cytokines that initiate their differentiation into a specific subset, each of which differs in their immune function, transcriptional activator, and the cytokine profile that is secreted in response to pathogens [130,137]. T helper 1 (T_H1_) cells are often activated in response to viral pathogens due to the high concentrations of activating cytokines, IL-12 and IFNγ, that are secreted by infiltrating APCs at the viral infection site. Upon their differentiation, T_H1_ cells provide a major source of IFNγ that drives the activation of APCs to phagocytose and inactivate HSV-1 and promote downstream CD8^+^ T cell activation [138,139,140]. High levels of IFNγ and IL-2, which are secreted by activated T_H1_ cells, were found in collected HSV-1-infected cornea tissues and are thought to contribute to viral clearance [141]. Additionally, HSV-1-infected corneas contained cytokines related to T helper 2 (T_H2_) cells during the later stages of infection. This observation likely indicates T_H2_ cell participation in late-stage inflammation and healing processes [141], as CD4^+^ T cells are implicated in the protection against corneal scarring after HSV-1 infection clearance [142]. Lastly, T follicular helper (T_FH_) cells, which mediate B cell interactions, were found to be activated in response to skin infections with HSV-1. While T_FH_ cells persist at the infection site after viral clearance, their migratory potential is speculated to be limited [143].

### 3.3. Adaptive Immune Responses to Latent Infection and Reactivation

The adaptive immune response and associated immune surveillance primarily play bystander roles during HSV-1 latent infection. The lack of viral replication and low-level presentation of antigens from latently infected cells prevents the direct clearance of HSV-1 from latently infected neurons by T cells. Despite the absence of a productive infection, innate and HSV-1-specific adaptive immune cells persist in the TG and are found clustered around latently infected sensory neurons [144]. While both HSV-1-specific CD4^+^ and CD8^+^ T cells are found in the TG, CD8^+^ T cells are the dominant adaptive immune cell positioned to act against HSV-1 during latent infection [138]. In the lip scarification model of HSV-1 infection, immune cell infiltration into the TG was observed as early as 2 days PI, with T cells also detected around latently infected neurons at 15 days PI [136]. Other murine models of HSV-1 infection have shown that the infiltration of CD8^+^ T cells into the TG peaks around 10–12 days PI and persists up to 90 days PI [145,146]. The infiltration of CD8^+^ T cells into the TG likely correlates with the increasing establishment of latent infection to suppress spontaneous reactivation and prevent the dissemination of HSV-1 into CNS tissues to cause disease. Despite their infiltration, the presence of CD8^+^ T cells does not seem to be cytotoxic to the latently infected sensory neurons [147].

#### 3.3.1. Effects of Immune Responses on Established Reservoirs in Innervating Neurons

The mostly transcriptionally dormant state maintained in sensory neurons during HSV-1 latent infection is mediated by the balance of LAT expression in latently infected neurons and the surrounding T cells in the TG [49]. LAT maintains latency through the transcription of microRNAs that actively suppress neuronal apoptotic pathways [148] and promote heterochromatin formation over replication-associated genes [149]. Despite the absence of broad viral gene expression and productive infection, T cells actively cluster around latently infected neurons for the duration of latent infection and actively suppress the reactivation of latent HSV-1 [144,147]. While the persistent activation of CD8^+^ T cells is not fully understood, some studies suggest that latently infected sensory neurons have low levels of viral gene expression, allowing continuous antigen stimulation and activation of the CD8^+^ T cells [144,150,151]. In support of this idea, immunogold electron microscopy detected MHC class I on sensory neurons juxtaposed to T cells in latently infected TG as early as 7 days PI [152], suggesting a T cell response mediated by viral epitope presentation in the context of MHC class I. These sensory neurons contained low antigen levels, suggesting a noncytolytic mechanism of responding CD8^+^ T cells that are not cytotoxic to neurons [152,153].

The HSV-1-specific CD8^+^ T cells present in the TG during latent infection share a similar phenotype to tissue resident memory T (T_RM_) cells, which may explain their persistence during latent infection [154,155,156,157]. This memory T cell differs from both central memory T (T_CM_) cells, which circulate between the blood and the spleen, and effector memory T (T_EM_) cells that circulate between the blood and non-lymphoid tissues. Instead, T_RM_ cells are localized to non-lymphoid tissues and respond to recurrent infections that occur in a specific tissue. Therefore, HSV-1-specific CD8^+^ T cells are speculated to be resident to the TG and are not replenished by acute infection in the periphery [158]. Their activation in their resident tissue is likely mediated by a tripartite interaction with CD4^+^ T cells, which are present during latent infection, and DC [159]. In the brain, CD8^+^ T_RM_ cells express the canonical markers CD103, CD69, and CXCR6, consistent with lower differentiation markers with increased PD-1 and CTLA-4, for a more regulated immune response in the resident tissue [154,160]. In response to recurrent pathogens, like HSV-1 in the TG, the CD8^+^ T_RM_ cells elicit noncytolytic responses as previously described, using IFNγ and granzyme B to prevent the spontaneous reactivation of HSV-1 in latently infected sensory neurons [156,161]. The expression of these molecules (i.e., IFNγ, granzyme B, TNFα, IL-1β) is further supported by the presence of innate immune cells, such as macrophages and DC, that are present in the TG and can further aid CD8^+^ T_RM_ cells in mediating the immune control of latent HSV-1 [162]. However, due to the persistence of these various immune cells and the continuous secretion of these molecules, HSV-1 infection can often result in chronic inflammation in the TG, which can contribute to inflammatory disorders in some individuals [163].

Despite an incomplete understanding about how CD8^+^ T cells are recruited to and are activated by latently infected neurons, they have been shown to be critical in preventing spontaneous reactivation in latently infected sensory neurons. T cells collected from the TG of mice were found to recognize HSV-1 ICP6 and VP16 proteins, which are expressed during reactivation [144]. Despite antigen recognition during latent infection, the amount of infectious virus in sensory neurons is low, and the lack of neuronal damage suggests a noncytolytic mechanism for surveilling CD8^+^ T cells [145,153]. This mechanism is speculated to be mediated by the secretion of IFNγ by CD8^+^ T cells, which was found in ganglionic neurons from HSV-1-infected rats [164]. IFNγ signaling was found to prevent reactivation through the inhibition of viral gene expression in latently infected neurons [165]. CD8^+^ T cells in the TG also express granzyme B that degrades viral gene products, like ICP4, implicated in reactivation [166,167]. Even though both IFNγ and granzyme B are inducers of apoptotic pathways, this response is blocked in latently infected neurons through the expression of LAT [168].

Other adaptive immune cells are present in the TG of latently infected individuals, but their roles are less described in comparison to CD8^+^ T cells. Like CD8^+^ T cells, CD4^+^ T cells cluster around latently infected sensory neurons in the TG and recognize select HSV-1 antigens [144]. However, their roles in maintaining latency or in suppressing reactivation are not fully understood. Since HSV-1 is primarily intercellular during latent infection and sensory neurons are not lysed by CD8^+^ T cells, the presentation of an exogenous antigen is unlikely despite many neural cells expressing MHC class II molecules [169]. While HSV-1-specific memory B cells are also found near latently infected sensory neurons, the exact role of humoral immunity in controlling reactivation is not clearly defined for HSV-1 infection [170].

#### 3.3.2. Adaptive Immune Responses Do Not Result in the Clearance of Latently Infected Neurons

Despite the importance of immunological control over HSV-1 infection, the host immune system is incapable of clearing HSV-1. This limitation of the host immune system is, in part, due to the novel immune evasion mechanisms of HSV-1 employed to escape immune surveillance and clearance of infected cells. Many of these evasion tactics are mediated by the tegument proteins contained in HSV-1’s structure and the viral genes expressed during acute infection [171,172]. For example, multiple HSV-1 proteins have roles in interfering with TLR-mediating signaling, blocking the activation of NF-κB and IRF-3 expression downstream. This inhibition leads to the reduction of pro-inflammatory and type I IFN expression in response to HSV-1 [19,173,174,175]. HSV-1 also hijacks the oxidative stress response as a novel evasion mechanism to further divert immune signaling pathways and promote its pathogenesis in the infected cell [62].

However, the primary reason the host immune response is incapable of clearing the infection is its inability prevent the establishment of latency in the TG, which allows the virus to hide from immune surveillance. However, this persistence leads to a chronic adaptive immune response, which, in return, can result in the T cell exhaustion of HSV-1-specific CD8^+^ T cells. T cell exhaustion is a hallmark of many chronic viral infections and is characterized by reduced effector function, poor proliferative capacity, and poor memory differentiation [176,177]. Exhaustion is often a result of consistent TCR engagement with antigens during chronic infections. Over time and as exhaustion progresses, CD8^+^ T cells may produce less IL-2, TNFα, and IFNγ [178,179]. Some studies suggest that the loss of T cell effector function is an immune evasion strategy influenced by long-term LAT expression. In a murine infection model, latency established by a HSV-1 LAT^+^ strain exhibited more functionally exhausted CD8^+^ T cells in the TG, as measured by IFNγ and TNFα expression, compared to mice infected with a LAT-null HSV-1 strain [180]. Additionally, exhausted HSV-1-specific CD8^+^ T cells exhibited higher levels of the inhibitory molecules PD-1 and Tim-3, which may confer reduced effector function [181]. A recent study also suggests that IFNα may play a role in latency establishment and T cell exhaustion, as its absence resulted in restored CD8^+^ T cell function and reduced latent reservoirs in the TG [182].

#### 3.3.3. Effects of Adaptive Immune Responses on Virus Reactivation

While the role of the adaptive immune response during latent infection is to suppress reactivation, many patients nevertheless experience recurrent outbreaks of acute infection. Due to the continuous and close association of CD8^+^ T cells and latently infected sensory neurons in the TG, it is understood that a disruption in the adaptive immune response is necessary for HSV-1 neuronal escape and reactivation [183]. While this disruption is likely triggered by stress stimuli [8], reactivation is not predictable and can be affected by the competence of the host immune system. In a 60-day cohort study, immunocompetent adults experienced an average of 1.4 reactivations, with a range of 0–30 reactivations per 30 days and 16.4 reactivations annually [184]. In contrast, reactivations in immunosuppressed individuals may be more frequent due to lower immunological control over infection [185].

##### In the Nervous System

Reactivation of HSV-1 from latently infected neurons in the TG is thought to be mediated by a balance of factors. As previously described, stress stimuli play roles in triggering reactivation events that can lead to the activation of the JNK pathway [45] and translocation of the GR receptor [46,47] for the remodeling of chromatin and subsequent lytic gene expression in sensory neurons. However, corticosteroids, often released during stress exposure, can dampen the immune response and lead to anti-inflammatory responses that disrupt immunological control over latent infection [186]. Stress associated with increased GR activity was also shown to enhance the function of T regulatory (T_reg_) cells by suppressing the immunosurveillance of TG-resident CD8^+^ T cells. With diminished CD8^+^ T cell responses in the TG, HSV-1 is able to reactivate and re-enter the periphery without being cleared by the immune system [187].

Immune suppression and compromised immune responses to reactivation in the TG can result from stress. HSV-1, as a viral pathogen that induces oxidative stress, was shown to upregulate T cell exhaustion phenotypes that lead to reduced effector function [180]. Stress can also impact the CD8^+^ T cell response against acute infection [188]. In the context of reactivation, stress stimuli can trigger the reactivation of latent reservoirs while also disrupting immunological control in the TG. In a murine model of psychological stress-induced reactivation, TG-resident CD8^+^ T cells exhibited a loss in effector function, which correlated with increased virus reactivation [189]. Similarly, female sex hormones, estrogen and progesterone, were able to induce the reactivation of HSV-1 by inhibiting TG-resident CD8^+^ T cell responses [190]. In a rabbit model, TG-resident CD8^+^ T cells following reactivation displayed markers similar to T cell exhausted phenotypes with reduced effector function, in comparison to CD8^+^ T cells isolated from latently infected rabbits. This phenotype was restored through the blockade of immune checkpoints, PD-1 and LAG-3, that are upregulated during T cell exhaustion [191]. In addition, there is an HSV-1-specific countermeasure that is detrimental to host immune responses. HSV-1 protein ICP47 promotes neurovirulence by way of blocking protection conferred by TG-resident CD8^+^ T cells [192].

##### In the Recurrent Lesion Caused by Reactivated Productive Infection

Immunological control of recurrent viral infections outside the latently infected nervous system is mediated by effector memory T cells that persist in nonlymphoid tissues. For example, intravaginal vaccination of human papillomavirus (HPV) in mice induced a memory T cell response that localized and persisted in the cervicovaginal tissue after primary and booster immunizations [193]. These cells are described as T_RM_ cells that are nonmigratory and persist in nonlymphoid tissues for up to six months at a time, providing rapid and long-term protection against recurrent infections [193,194,195]. After the initial exposure to the antigen, T_RM_ cells are estimated to make up 98% of the effector memory T cells present in the skin [196].

T_RM_ cells are also involved in immune responses to reactivated epithelial infections in herpes labialis. Following reactivation of HSV-1 from the TG, HSV-1 re-enters the mucosa from the TG axon that innervates the initial site of transmission. Re-exposure of HSV-1 at the mucosa was found to stimulate a persistent memory CD8^+^ T cell response that is resident to the acute infection site [156]. In HSV-1 mouse models, these CD8^+^ T_RM_ cells are localized at the sensory nerve endings at the point of entry for HSV-1 into the mucosa [197]. The infiltration of virus-specific nonlymphoid tissue resident T_H1_ cells [198] and memory B cells [199] was also demonstrated following primary viral infections. In HSV-2-immunized mice, infiltrating memory T_H1_ cells were shown to mediate antiviral activity through the secretion of IFNγ to limit the dissemination of the virus [200].

Humoral responses specific to HSV-1 appear to have less effect on recurrent infection. Memory B cells and subsequent HSV-1-specific antibody responses are understood to be limited during HSV-1 primary and recurrent infection. While local HSV-1-specific antibody titers in the lesion increase during reactivation, changes in serum HSV-1-specific antibody titers are insignificant [170,199].

## 4. Current Immunological Interventions for Treating Herpes Labialis

There are currently no immunological therapeutic approaches for either preventing HSV-1 infection or treating an existing infection. Although standard of care (SOC) drugs are available and approved to treat established HSV-1 infections, they add nothing to the host immune responses to HSV-1 infection and do not serve to cure the infection since they have no effect on long-term virus persistence in latently infected neurons. In addition, more recent studies have revealed that pharmacological treatments of infection may have detrimental effects on host immune responses to HSV-1. Furthermore, there are currently no HSV-1-specific vaccines approved for use in a prophylactic or therapeutic capacity, despite many historical successes in the development of vaccines against viral pathogens.

### 4.1. Standard of Care Pharmaceutical Options May Be Detrimental to Host Immune Responses to HSV-1 Infection

SOC therapies for oral HSV-1 infection are nucleoside analogs, with acyclovir (ACV) and ACV derivatives being the most common, that can be administered either topically or orally depending on the patient and infection severity. ACV and its derivates (penciclovir, valacyclovir, and famciclovir) are preferentially taken up by HSV-1-infected cells that express the viral thymidine kinase (TK) [201]. ACV undergoes phosphorylation by the viral TK and subsequent phosphorylation events by cellular kinases to generate a triphosphate derivative that acts as a competitive inhibitor of the viral DNA polymerase. This active form of ACV inserts itself into the elongating viral DNA, thereby terminating replication. Thus, the assembly and release of progeny virions from the infected cell are inhibited [202]. Despite their inability to clear HSV-1, nucleoside analogs are considered SOC due to their high safety profiles [203,204] and clinical efficacies. With oral and topical formation, ACV was shown to alleviate pain, reduce lesion size, and minimize viral shedding at the acute infection site, leading to a shortened symptomatic infection [205,206,207,208]. This efficacy can further be improved with the ACV derivates, which allow shorter treatment regimens by offering increased bioavailability and longer half-lives [209].

The antiviral effects of SOC therapies are temporary with intermittent use. While SOC drugs can shorten the symptomatic window of acute infection and limit viral seeding into the TG, they are ineffective with respect to fully preventing the establishment of persistent latent infection. Furthermore, SOC therapies have no effect on latently infected sensory neurons where viral replication is absent, and they have minimal effects on reactivation [210]. To achieve long-term benefits from SOC therapies, infected individuals are required to take oral ACV daily over extended periods of time, with the goal of reducing the number of symptomatic outbreaks. This was validated in patients with genital HSV-2, where continuous oral ACV therapy reduced outbreaks of symptomatic infection to 1.4–1.9 per year, compared to 7.0–12.6 outbreaks in individuals on intermittent ACV therapy [203]. Another study reported that 61% of patients on long-term oral ACV therapy were recurrence-free after 3 years, with no drug toxicity [211]. While tong-term treatment can reduce outbreaks, those positive effects cannot be achieved with the intermittent usage of ACV [212].

Relevant to the immunological control of HSV-1 infection, long-term antiviral therapy may also have effects that are counter to the intended control of HSV-1 recurrence. A small number of studies of genital herpes caused by HSV-2 and infections by other herpesviruses, such as cytomegalovirus (CMV), have indicated associations between long-term ACV therapy and immunosuppression. In patients infected with CMV, T cell reactivity toward CMV-specific viral proteins was reduced by 53% after one year of continuous low-dose ACV therapy [213]. Similarly, the antiviral humoral immune response was found to be compromised in HSV-2-infected patients after one year of daily ACV therapy, as measured by reduced serum HSV-2-specific IgG. In contrast, patients on short-term ACV therapy had no changes to their serum antibody titers [214]. Earlier studies also found that lower antibody production was correlated with reduced lymphocyte proliferation in response to HSV antigens in ACV-treated patients [215]. Because patients with HSV-1 primarily use ACV intermittently, the adverse effects of ACV on the HSV-1-specific immune response may be smaller but still impactful. Given the potential differences in immune response dynamics during HSV-1, HSV-2, and CMV infections, a complete understanding of the effects of ACV therapy on host immune responses during herpes labialis will require further study.

Mouse models of infection appear to corroborate observations made in humans. Mice with cutaneous HSV-1 infection treated with long-term and early ACV had lower HSV-1-specific IgG production, despite faster resolution of lesions. In contrast, mice that received shorter and late ACV treatment had higher antibody production but more severe viral lesions. While the correlation between increased efficacy and weakened humoral responses is unclear, this study demonstrates the modification of the immune response by ACV [216]. Given the importance of immunological control over HSV-1 infection, modified immune responses resulting from antiviral drug therapy may be contrary to the goals of pharmaceutical treatments.

The long-term use of ACV is also associated with emerging antiviral resistance, which will reduce the efficacy of SOC drugs and potentially increase the virulence of HSV-1. ACV-resistant strains are often isolated from immunocompromised individuals due to their proclivity for long-term ACV therapy and more frequent reoccurrences of symptomatic infection. For immunocompromised individuals, ACV resistance develops at a rate of 3.5 to 10% of the population, with some surveys reporting rates as high as 36%. While the rates for immunocompetent individuals are considerably lower, resistance can still emerge [202].

### 4.2. Lessons Learned from Vaccines Under Development for HSV-1 Infection

The high global prevalence and burden of a lifelong infection emphasize the need for effective preventative strategies, like vaccines, against HSV-1. For many viral infections, vaccines are an effective antiviral preventative strategy that limit the spread of pathogens in the population through the establishment of herd immunity. For prophylactic vaccines, the goal is to establish an immunological memory response against a specific antigen, or multiple antigens, in a naïve host. The immune response can be humoral or cell-mediated, mediating a rapid immune response upon exposure to the pathogen [217]. However, the development of prophylactic vaccines against HSV-1 have been challenged by the failure to stimulate an adequate HSV-1-specific immune response capable of preventing or clearing HSV-1.

Many types of vaccination strategies have been tested that rely on the production of a robust HSV-1-specific neutralizing antibody responses against viral glycoproteins that are typically the immunodominant peptides for HSV-1-specific lymphocytes [12,116,131]. HSV-1-specific antibodies alone have proven to be insufficient in preventing acute or latent infection in individuals exposed to HSV-1 [49,170]. This was shown with two subunit vaccines (gB/gD/MF59 and gD-2/AS04), which yielded protection in animal models but ultimately failed to establish a neutralizing antibody response in phase 3 clinical trials [218]. It is also possible that targeting viral glycoproteins alone is not sufficient in stimulating immune control over HSV-1. In contrast, the GEN-003 subunit vaccine containing HSV-2 gB, ICP4, and M2 conferred immunological protection in phase III clinical trials. Protection was shown to be mediated by the stimulation of both humoral and cell-mediated responses, highlighting the importance of a HSV-1-specific cell-mediated response [219].

Immune responses to herpesviruses other than HSV-1 may also provide immunological protection against HSV-1. While the candidate vaccine GEN-003 was evaluated for protection against HSV-2 infection, recent studies suggest that HSV-2 glycoproteins may offer protection against HSV-1. This was demonstrated in an mRNA trivalent vaccine containing HSV-2 gC, gD, and gE antigens [220]. Preclinical studies have demonstrated a robust humoral immune response in preventing genital disease and reducing the establishment of latency in the dorsal root ganglia in 100% and 97% of mice, respectively [221,222]. However, previous vaccine clinical trials have shown that neutralizing antibodies do not elicit complete immune control over infection. Given the use of HSV-2 antigens, it is possible that the mRNA trivalent vaccine may not offer complete protection against HSV-1. Additionally, the mRNA trivalent vaccine is prophylactic, with no indication that it can clear or affect established latent reservoirs. Given the prevalence of HSV-1 in the population and the high incidence of early-life infection, this vaccine alone will not be enough to reduce the global burden of HSV-1.

Live attenuated vaccines have also been explored for HSV-1 and are capable of inducing both humoral and cell-mediated memory responses. One candidate, HSV-1 0∆NLS, was tested as a prophylactic vaccine against ocular HSV-1 infection, establishing a T cell-dependent humoral response capable of preventing ocular pathology and subsequent dissemination into neural tissues in mice [223]. Specifically, this vaccine was capable of producing specific antibody responses against various HSV-1 antigens in the envelope, tegument layer, and capsid [224]. However, this immune protection against ocular HSV-1 infection diminished over time, suggesting a limited capacity for live attenuated vaccines in stimulating a robust protective response [225]. The live attenuated VC2 vaccine was also successful in protecting mice from HSV-1-induced ocular disease [226]. Similarly, the R2 vaccine, which contains a mutation in the HSV-1 pUL37 tegument protein involved in viral egress, prevented HSV-1 from establishing a latent infection in the guinea pig TG [227]. While shown to be effective in preventing disease in preclinical models, live attenuated vaccines have historically questionable safety profiles, making them unlikely vaccine candidates [228].

Prophylactic vaccine strategies against HSV-1 have faced many challenges that have prevented their success. This lack of success is partly due to the target memory immune response, with many strategies often relying on the stimulation of a protective neutralizing antibody response. Not only have humoral responses not been implicated in immunological control over HSV-1, but prevention of HSV-1 infection requires a high antibody titer. While this may not be a challenge in preclinical animal models, antibody titers in humans decline over time and may not be at a suitable concentration when the immunized individuals become re-exposed to HSV-1. Greater success in prophylactic vaccination strategies have been achieved when the stimulation of a cell-mediated response is targeted, as HSV-1-specific T cells have a more dominant role in exerting immune control over HSV-1 acute and latent infection. The high global infection rate of HSV-1 also poses a challenge for developing prophylactic vaccines. Currently, ~70% of the world’s population under the age of 50 harbors a HSV-1 infection, with the majority becoming infected early in life. This high and early infection rate makes prophylactic vaccines obsolete for the majority of the population. Given its global prevalence, therapeutic vaccination strategies or immunotherapies would be better to target HSV-1 infection. Instead of stimulating a response, these vaccines should be focused on enhancing pre-existing immunological control, as the host immune system is incapable of clearing HSV-1 reservoirs or suppressing reactivation.

## 5. Non-Thermal Plasma as the Basis for a Novel Antiviral and Immunological Therapy for Herpes Labialis

Non-thermal plasma (NTP) is a non-invasive therapeutic technology that has established antiviral and immunomodulatory activities. NTP is partially ionized gas composed of chemical, radiative, and thermal components that collectively contribute to its biological activity; principle among them are the short- and long-lived reactive oxygen and nitrogen species (RONS) it delivers to biological targets. NTP is shown to inactivate pathogens, modulate oxidative stress responses, induce cellular signaling pathways, stimulate proliferation, and enhance immunogenicity in cells [229,230,231]. We recently reviewed the potential of NTP to overcome many of the challenges associated with the current therapies and new vaccine strategies to be used in cure strategies against oral HSV-1 infection [62].

NTP inactivates many types of viral pathogens [232]. For decontamination, NTP is a safe and environmentally friendly tool that can inactivate contaminants found in drinking water [233] and foodborne pathogens like feline calicivirus (FCV), a surrogate for human norovirus [234]. The potent antiviral activity of NTP is often correlated with RONS concentrations, which are easily controlled by changing generation power and treatment durations. The application of NTP, particularly with higher power and longer exposure, produces high concentrations of RONS, which modify the structural components of viruses, such as the FCV capsid, leading to inactivation [234,235]. Similarly, the treatment of hepatitis B virus (HBV)-contaminated blood with NTP resulted in viral inactivation due to the oxidation of HBV antigens [236].

Until recently, most studies were limited to assessments of NTP as an agent of disinfection. NTP was applied to cell-free viruses on an inanimate surface or in an aqueous environment to model, respectively, NTP application to virus-contaminated surfaces (i.e., fomites like table tops or doorknobs) or liquids (i.e., wastewater effluents or bodily fluids). However, very few studies have examined the efficacy of NTP to disrupt the infection cycle when applied to virus-infected cells or tissues. The goal of our studies has been the exploration and development of therapies for viral infections that leverage NTP as an agent with antiviral and immunomodulatory activities [237,238].

### 5.1. NTP Has Multiple Effects on HSV-1 Infection and Replication

NTP is a promising antiviral therapy alternative that can impact multiple stages of the HSV-1 replication cycle in the treated lesion. We have recently demonstrated the direct antiviral effects of NTP using an in vitro model of HSV-1 replication in cold sores. Specifically, NTP application inactivated cell-free HSV-1, disrupted viral replication in HSV-1-infected keratinocytes, and prevented HSV-1 infection in keratinocytes pre-exposed to NTP at doses that were non-toxic to the mammalian host cell [239]. These observations suggest that NTP acts as an antiviral agent at different stages of the HSV-1 replication cycle and may be more effective than SOC drugs.

#### 5.1.1. NTP Disrupts HSV-1 Replication in Infected Cells

While a limited number of in vitro studies have shown that NTP application disrupts, and potentially shuts down, viral replication in HSV-1-infected cells during acute infection, the exact mechanism is unknown [239,240]. One potential mechanism could be through RONS-mediated modification of nucleic acids in the virus. Since viruses do not possess DNA repair mechanisms like host cells, damage induced by RONS could disrupt synthesis by the viral DNA polymerase. NTP was shown to induce DNA damage markers, 8-OHdG [241] and γ-H2AX [242], in mammalian cell lines; specifically, γ-H2AX was mediated by intracellular RONS induced by NTP [242]. It is also possible that NTP RONS permeate the cell membrane and intracellular compartments, gaining access to the viral genome. In particular, this is found to be true for less hydrophilic RONS, such as nitric oxide, ozone, nitrogen dioxide, and dinitrogen tetroxide, that can cross cell membranes [243]. Together, the damage to the viral genome by NTP RONS may prevent the replication and assembly of progeny virions inside cells undergoing HSV-1 replication. In a scenario in which NTP is applied as a treatment for oral herpes, the detrimental effect of NTP on HSV-1 replication in infected epithelial cells could manifest as a reduced viral burden in the lesion site, limiting the spread of infection and enhancing the resolution of cold sores. Importantly, because NTP targets multiple steps in the HSV-1 lifecycle, the likelihood of resistance to this therapy is low, a major concern for SOC therapies.

#### 5.1.2. NTP Reduces Keratinocyte Susceptibility to HSV-1 Infection

NTP application also reduced keratinocyte susceptibility to HSV-1 infection [239]. This reduced susceptibility could be indicative of the effect of NTP on the initial stages of HSV-1 infection by blocking viral attachment and entry at the target cell surface. Similar effects were reported in pre-treated macrophages that exhibited reduced HIV-1 membrane fusion through the modification of the cell surface molecules (CD4 and CCR5) necessary for HIV-1 binding and entry [244]. In an in vitro model for gingival tissues, proteoglycans were found to be highly susceptible to RONS-mediated modifications following exposure to both hydrogen peroxide and hydroxyl radical [245], both of which are produced in NTP [62]. NTP was also shown to promote the temporary internalization of ACE-2 entry receptors in cell lines permissible for SARS-CoV-2 [246]. We propose a similar preventative antiviral mechanism for the HSV-1 infection of keratinocytes, wherein heparan sulfate, the entry receptor for HSV-1, is modified by NTP. Heparan sulfate is composed of a core protein attached to repeating disaccharide units, which interact with the HSV-1 envelope glycoproteins [247]. Modification of heparan sulfate by NTP could be a mechanism by which HSV-1 binding is compromised, protecting the cell from infection, as demonstrated by reduced HSV-1-infected cells 24 h post-NTP [239]. This finding is indicative of increased membrane turnover, changing the abundance of cell receptors on target cell membranes and making it more difficult for HSV-1 to initiate viral attachment. Overall, these findings indicate that the extracellular molecules involved in entry could be modified by NTP, preventing viral attachment and entry into pre-treated cells.

Taken together, the observed in vitro effects of NTP on the HSV-1 replication cycle suggest that NTP could be an efficacious therapy for cold sores that could reduce the viral burden and enhance the resolution of lesions better than SOC therapies. Many of the limitations of SOC therapies for HSV-1 can be addressed by NTP. NTP not only disrupts viral replication but also blocks the infection of naïve cells for the enhanced resolution of cold sores. Further, it may indirectly affect viral seeding into the TG. The application of NTP to a lesion where sensory nerve endings also become exposed to NTP may alter the susceptibility of sensory neurons to HSV-1 entry.

### 5.2. NTP as a Multi-Functional Therapy for HSV-1 Infection

NTP application as a therapy for herpes labialis could be efficacious due to its capacity to disrupt multiple aspects of infection, replication, and the spread of the virus. The reduction of virion infectivity, the disruption of virus replication, and the reduction of cell susceptibility to infection could combine to reduce viral pathogenesis as well as the spread of the virus locally and to the TG. In addition, the immunomodulatory activity of NTP may have immediate and long-term benefits in an HSV-1-infected individual.

#### 5.2.1. NTP Antiviral Activities Will Limit Virus Spread and Pathogenesis in the Lesion

In an envisioned NTP-based treatment for herpes labialis, NTP would be applied locally to the herpetic lesion of infected symptomatic individuals. Here, NTP would target cell-free virus, infected cells and the surrounding uninfected keratinocytes. While NTP is capable of inactivating cell-free HSV-1 [241,248], NTP has also been shown to disrupt HSV-1 replication in vitro in explanted HSV-1-infected human corneal cells [240] and in HSV-1-infected human keratinocytes [239]. Disruption of HSV-1 replication in the lesion would reduce infectious virus production and, therefore, reduce HSV-1 propagation and spread within the local lesion. The benefits of this effect would be diminished pain and inflammation associated with the infection, as well as reductions in the time to lesion resolution.

#### 5.2.2. Potential Effects of NTP on the HSV-1 Latent Reservoir in Neurons

An additional benefit of NTP-mediated reductions in HSV-1 replication, production, and spread in the treated lesion could be its effects on the establishment of latent infection in the TG, an aspect that has not yet been investigated. The accelerated resolution of acute infection after NTP could reduce viral seeding into the TG and lead to fewer established latent reservoirs. While the neuron bodies of the TG are distal from the lesion and are not subject to the direct effects of NTP, the nerve endings of sensory neurons that innervate the site of infection are likely to be affected by NTP since they are near the cell surface where NTP will be applied. Sensory nerve endings proximal to the site of NTP application could have reduced susceptibility to infection by HSV-1, limiting viral seeding into the TG. Like keratinocytes, this reduced susceptibility to infection could be caused by the modification of the entry receptors of sensory neurons, heparan sulfate and nectin-1, resulting in compromised attachment to and entry of the virus into the nerve ending. However, the effects of NTP on HSV-1 attachment to and entry into sensory neurons remain to be studied. Similarly, our hypothesis that NTP treatment of cold sores results in reduced viral seeding in the TG needs to be tested in future studies.

The NTP treatment of axons may also trigger neuronal signaling pathways that could impact latency establishment. In a rat model for sciatic nerve crush injury, NTP treatment was demonstrated to promote healing responses in damaged neurons, restoring axon fibers and the myelin sheath [249], likely mediated through the stimulation of neuronal signaling pathways. Human neuroblastoma cells exposed to NTP displayed markedly increased neuron proliferation markers, wnt3 and b-catenin, that contribute to axonal elongation while preventing the overexpression of neuroma markers [250]. NTP was also shown to induce neuronal differentiation through the activation of the Trk/Ras/ERK signaling pathway [251].

Stimulation of these signaling pathways in neurons by NTP may also be relevant to the application of NTP to an existing infection in which a latent infection has already been established. During latent infection, LAT modulates many neuronal signaling pathways critical for maintaining latency. One such pathway, the JNK pathway, is suppressed by LAT but becomes activated as a mechanism of reactivation [45]. The JNK pathway is also activated by NTP, mediating tumor cell killing [252]. NTP could similarly induce neuronal cell pathways in latently infected neurons, impacting the ability of HSV-1 to establish and maintain latency. This potential effect of NTP on neurons needs investigation.

#### 5.2.3. NTP-Associated Immunomodulation During HSV-1 Infection

NTP stimulates antiviral immune responses and enhances immunogenicity in virus-infected cells, as shown in an in vitro model for latent HIV-1 infection [238]. We postulate that NTP will also induce modulation of the immune response to HSV-1. Since the host immune system is incapable of clearing HSV-1, boosted antiviral immunity following NTP treatment could enhance the immune targeting of HSV-1-infected cells and suppress reactivation from the TG.

NTP is proposed as a potential immunotherapy for oral HSV-1 infection that will exert greater immunological control over acute, latent, and recurrent infection. The immunomodulatory impact of NTP has been primarily studied in in vitro and in vivo models of cancer, as the host’s adaptive immune cells are often exhausted and ineffective in clearing malignant cells in tumor sites [253]. Similar to cancer, the lifelong persistence of HSV-1 is aided by the inefficiencies in the adaptive immune response in clearing latent reservoirs and the rise of T cell exhaustion over time [254]. The first report of immunomodulation after NTP was in an in vivo model of colorectal tumors, in which treatment promoted the release of damage-associated molecular patterns (DAMPs), subsequent infiltration of APCs into the tumor site, and activated tumor-specific CD8^+^ T cells [255]. We propose a similar immunomodulatory mechanism of NTP for oral HSV-1 infection. Given the limitations of the host immune system over infection, we believe that NTP can boost the existent HSV-1-specific CD8^+^ T cell response for greater control over infection.

Other disease and infection models have replicated this immunomodulatory effect of NTP, resulting in enhanced antigen-specific immune cell functions. In an in vitro model of pancreatic cancer, NTP induced the expression of immunostimulatory cytokines, promoting their phagocytosis of cancer cells by co-cultured DCs [256]. Enhanced DC function was also observed in a Lewis Lung cancer model in which DCs released increased proinflammatory cytokines, contributing to reduced tumor growth [257]. In the context of viral infections, the NTP exposure of Jurkat cells latently infected with HIV-1 (J-Lat cells) induced the surface presentation and release of DAMPs, promoting phagocytosis and the recruitment and maturation of macrophages [237,238]. Surface presentation of DAMPs like calreticulin (CRT) and heat shock proteins (HSPs) can serve as “eat-me” signals for phagocytes, while the release of ATP, high mobility group box protein 1 (HMGB1), and proinflammatory cytokines can aid in immune signaling and cell recruitment [258]. Furthermore, enhanced cell immunogenicity was evident in the altered arrays of viral antigens presented on MHC class I molecules during infection, suggesting a mechanism by which the course of the adaptive immune response may be altered by NTP treatment [238]. So far, immunomodulation by NTP in virally infected cells has only focused on innate immune responses, which are not antigen-specific and are general mechanisms used by a variety of eukaryotic cells. Therefore, we are confident that the innate immune responses observed in NTP-exposed J-Lat cells can be translated to HSV-1-infected cells after NTP exposure. Similarly, because the MHC class I presentation of antigens is a process carried out by all nucleated cells, NTP-associated changes in the T lymphocyte immunopeptidome support the hypothesis that similar changes will be seen in NTP-exposed epithelial cells infected with HSV-1.

For therapeutic benefit in HSV-1 infections, NTP is expected to stimulate local antiviral immune responses at the site of infection with the subsequent development of more robust and specific systemic adaptive immune responses. When DCs activated by NTP were injected into the peritoneal cavity of mice with Lewis Lung cancer, they promoted the infiltration of CD4^+^ and CD8^+^ T cells into the tumor microenvironment, boosting the adaptive immune response and reducing tumor growth. This was accompanied by the downregulated expression of PD-1 and STAT1, markers of functionally exhausted T cells [257]. These observations suggest that NTP could also restore T cell function. Since T cells display an exhausted phenotype in chronic HSV-1 infection models, the restoration of T cell function could enhance their role in clearing HSV-1-infected cells in the periphery and suppressing reactivation in the TG.

While the immune responses stimulated by NTP have been characterized in murine models, they may not fully reflect NTP-induced immunomodulation in humans. For HSV-1, herpes labialis has been modeled in mice using the lip scarification model of infection, which produces cold sores by manually scarifying the lower lip of mice with a needle [136]. Given the artificial nature of the virus inoculation, there may be differences in acute infection location, infection spread, and lesion microenvironment that differ from human herpes labialis. Furthermore, mice have discernable differences in their innate and adaptive immunity [259], which may be suggestive of the immunomodulatory effect of NTP but will challenge the characterization of antiviral immunity that can be induced by NTP treatment. Furthermore, the use of mouse models to study the NTP treatment of herpes labialis (particularly the lip scarification model) are ineffective in examining the effect on recurrent infections.

However, the utility of animal studies to demonstrate the efficacy of NTP therapy is suggested by human studies of NTP therapy. In two small clinical trials of NTP as a therapy for warts, NTP application resulted in the clearance of warts of presumed human papillomavirus (HPV) etiology [260,261]. These studies suggest the success of animal studies of NTP as a therapy for herpes labialis and the potential for demonstrating the underlying antiviral and immunological effects of NTP application in animals, as well as demonstrating the efficacy of an NTP-based treatment for HSV-1 infection.

As an immunotherapeutic strategy for HSV-1 infections, the topical application of NTP to HSV-1 cold sores could induce local antiviral immune cells, increase the breadth of antigens presented by DCs, and stimulate the development of a more robust HSV-1-CD8^+^ T cell response capable of exerting greater immune control over the establishment of viral latency and subsequent reactivation.

## 6. Conclusions and Future Directions

The course of HSV-1 infection in the absence of treatment is under the control of the host immune system. Antiviral immune responses are initiated during the first exposure to HSV-1, which causes a symptomatic acute infection and leads to the activation of a robust adaptive immune response. Specifically, HSV-1-specific CD8^+^ T cells mediate immune control over HSV-1 by directly targeting HSV-1-infected mucosal epithelial cells to resolve viral replication and establishing a tissue resident memory phenotype to control future outbreaks of acute infection. While CD8^+^ T cells are capable of resolving symptomatic infection, they cannot clear HSV-1 once it enters the TG to establish a latent infection and escape immune surveillance. Despite the absence of viral replication in the TG, the adaptive immune system remains active and clusters around latently infected sensory neurons. Instead of killing latently infected cells, HSV-1-specific CD8^+^ T cells elicit noncytolytic mechanisms to suppress reactivation and prevent the dissemination of HSV-1 into the nervous system. When this adaptive immune response is disrupted, or weakened, reactivation occurs and leads to outbreaks of symptomatic acute infection. Additionally, a weakened adaptive immune response can lead to the dissemination of HSV-1 and cause HSV-1-associated disease, highlighting the importance of a strong immune response.

Despite the robust anti-HSV-1 immune response, HSV-1 is not cleared from the host. Vaccination strategies to overcome the limitations of immune control have been unsuccessful. In this review, we make a case for NTP as a disruptive technology, which has both antiviral and immunomodulatory activity (Figure 3). It is an inexpensive, safe, and non-invasive strategy for the control of acute and latent HSV-1 disease. Our previous investigations demonstrated the antiviral efficacy of NTP in disrupting viral replication and reducing susceptibility to HSV-1 infection, both of which will contribute to accelerated lesion resolution and reduced virus seeding into the TG. Furthermore, we hypothesize that NTP will boost the host immune system by expanding the breadth of antigens presented to the adaptive immune system and the development of an HSV-1-specific CD8^+^ T cell response. By strengthening the CD8^+^ T cell response, which can be bypassed by HSV-1 as the virus reactivates and causes recurrent disease, NTP will facilitate a more robust host immune response that will exert greater immune control in limiting acute infection and suppressing the reactivation of latent infection.

The importance of understanding host immune responses against and therapies applied to HSV-1 infection is underscored by two important aspects of the current epidemiological picture of HSV-1. First, recrudescent oral HSV-1 infection is very prevalent in the global population. Second, HSV-1 is becoming more prevalent as a cause of genital herpes [262]. An understanding of host immune responses toward HSV-1 acute and latent infection will be critical in light of the shift in HSV-1 epidemiology from oral to genital transmission. What we learn about HSV-1 infection and disease will also augment our understanding of HSV-2 as a sexually transmitted pathogen. Furthermore, knowledge of the effectiveness of NTP as an antiviral and immunomodulatory agent used to treat HSV-1 infection will be readily applied to efforts to develop NTP as an additional weapon in the armamentarium used to treat HSV-2 genital herpes.

## Figures and Tables

**Figure 1 viruses-17-00600-f001:**
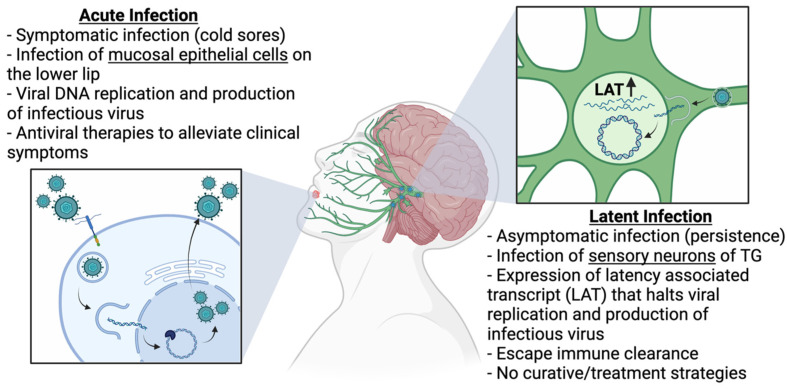
Overview of HSV-1 infection. During acute infection, HSV-1 replicates and produces the infectious virus in mucosal epithelial cells within the oral mucosa of infected individuals. This leads to the development of orofacial cold sores that can be treated with antiviral therapies to shorten the duration of symptomatic infection. At the same time, HSV-1 virions produced during acute infection seed the axons of the trigeminal ganglia (TG). These latently infected sensory neurons escape immune surveillance and clearance through silenced viral replication and virus production. Due to the lack of therapies that can affect latent infection, HSV-1 is incurable. As a consequence, infected individuals are potentially subjected to a lifetime of recurrent outbreaks of symptomatic infection.

**Figure 2 viruses-17-00600-f002:**
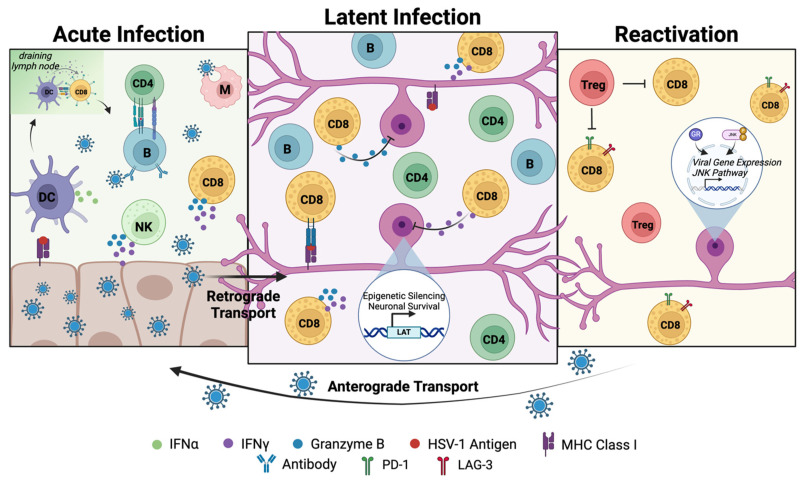
Anti-HSV-1 Immune Response Overview. HSV-1 infection is controlled by the host immune system. During acute infection, various innate immune cells are involved in innate killing (NK), phagocytosis (M, DC), and antigen presentation (DC) to activate the adaptive immune response to directly target HSV-1-infected cells (CD8) and mediate the humoral immune response (CD4). As HSV-1 establishes a latent infection, the adaptive immune response remains active and clusters around HSV-1-infected sensory neurons (CD8, CD4, B), despite the absence of viral replication, to actively suppress reactivation through noncytolytic mechanisms (CD8). When this adaptive immune response is disrupted or weakened by neuronal signaling, T_reg_ activity, or T cell exhaustion, viral gene expression is activated and HSV-1 undergoes anterograde transport back to resume acute infection.

**Figure 3 viruses-17-00600-f003:**
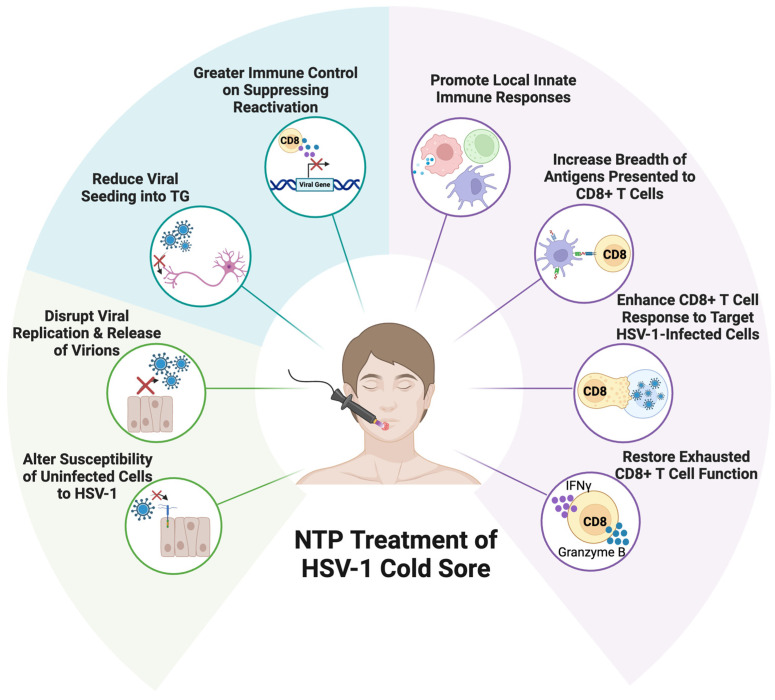
Overview of NTP as an Immunotherapy for Oral HSV-1 Infection. NTP treatment would elicit both direct and indirect antiviral and immunomodulatory effects on HSV-1 infection. Based on in vitro findings, NTP was shown to disrupt viral replication, reduce viral release, and alter the susceptibility of uninfected cells to infection with HSV-1. By limiting acute infection of the cold sore, this would reduce the quantity of virus that would seed the TG and establish a latent infection. Direct NTP treatment has also been shown to enhance immunogenicity and local cell responses in virally infected cells in vitro. By increasing the breadth of antigens presented by DCs, NTP would indirectly result in the stimulation of a more robust CD8^+^ T cell response that can more effectively target HSV-1-infected cells, restore exhausted T cell phenotypes, and maintain the suppression of latent infection.

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
