# Peer review of "Immunological Control of Herpes Simplex Virus Type 1 Infection: A Non-Thermal Plasma-Based Approach"

_viruses, 2025, doi:10.3390/v17050600_

Round 1
Reviewer 1 Report
Comments and Suggestions for Authors
Dear authors!
1) The abstract of your manuscript mentions that non-thermal plasma (NTP) has antiviral effects and stimulates local immune responses in vitro. Consequently, authors propose NTP as an immunotherapy. The discussion extends this to in vivo therapeutic effects without bridging studies. The leap from in vitro to in vivo without intermediate evidence is rather speculative.
2) The manuscript claims that long-term acyclovir (ACV) therapy might suppress the immune response (Section 4.1). However, the references cited (212, 213) mention cytomegalovirus and HSV-2. Using studies on different viruses to support ACV's effect on HSV-1 immune response could be inconsistent if the immune dynamics differ.
3) Section 5.1.1 (line 722) narrates that NTP disrupts HSV-1 replication, possibly via reactive oxygen and nitrogen species (RONS) damaging viral DNA. While RONS are implicated, the article admits the "exact mechanism is unknown" and conflates in vitro keratinocyte data with speculative in vivo effects. No direct proof of RONS penetrating neurons or targeting viral DNA in latency.
4) Section 5.1.2 suggests that NTP reduces cell susceptibility by modifying heparan sulfate, an HSV-1 receptor. However, in Section 5.2.2, authors hypothesize that NTP might affect sensory neuron entry receptors, but the evidence for this in neurons isn't provided. The in vitro studies were performed on keratinocytes, so extrapolating to neurons might be a stretch without data.
5) In Section 5.2.3, the authors suggest NTP enhances CD8+ T cell responses by increasing antigen presentation. However, the in vitro models mentioned (e.g., HIV-1 in J-Lat cells) might not be directly translated to HSV-1. The cited evidence for immunomodulation does not directly apply to HSV-1. The manuscript lacks data showing NTP-induced antigen presentation changes in HSV-1-infected cells.
6) NTP’s efficacy in animal models (e.g., murine HSV-1) supports human therapeutic potential.
While human immune responses and lesion microenvironments differ significantly from murine models. In this way presented article does not address translational challenges. It might be that authors can propose some experiments supporting this hypothesis.
Some of the presented issues could be addressed. The authors do mention some limitations, which could possibly be extended.
If further investigation is needed, then it should be discussed in a positive way, but few key claims lack direct evidences.
Author Response
Comments and responses have been provided in the attached Word document.

Reviewer 2 Report
Comments and Suggestions for Authors
Sutter et al describe much in detail the immunological control of HSV 1 infection.
In addition, a non-thermal plasma-based approach to treat HSV infections is proposed.
- Sutter et al stress the infection of HSV 1 as previously (1) but describe HSV infection in general in some sections. HSV 2 is a much larger problem in particular when recurrent HSV infections are described. The authors should clarify as to why they specifically target HSV 1. The reviewer may assume that the non-thermal plasma-based approach the authors suggest would only be of some effect for HSV 1 infections.
- As the mouse is one of several model system that mimic some of the symptoms experienced in humans. Non human studies should be clearly marked.
- For the clinical aspect of mucosal infections in humans and recurrent infections the clinical description is minimal. Without being exclusive but important clinical work from Schiffer and Corey is mentioned in a single 1984 paper in association with acyclovir.
- The latent infection and reactivation of HSV is extensively described which possibly contributes to its importance. The authors mention T cell exhaustion shown in mice in studies with LAT antigens. In human studies CD8 T cells against ICP6 and VP16 antigens were described. Do the authors know of or have data how the antigen is fed to the MHC I molecules to be detected by the T cells. Do exosomes and possible other ways of antigen transfers play a role? Although B cells present in ganglia are described but macrophages and dendritic cells among others are also present in ganglia. What is the role of these cells?
- Non-thermal plasma-based approaches have been described in vitro in cell cultures inhibiting HSV replication to some degree. Natural oxidative stress responses are present in most all cells. But how could a non-thermal plasma-based approach really have an effect on the real burden of HSV infection – (mostly) HSV 2 in recurrent infections.
Reference
- Sutter J, Bruggeman PJ, Wigdahl B, Krebs FC, Miller V. 2023. Manipulation of Oxidative Stress Responses by Non-Thermal Plasma to Treat Herpes Simplex Virus Type 1 Infection and Disease. Int J Mol Sci 24.
Author Response
Reviewer comments and our responses have been provided in the attached Word document.

Round 2
Reviewer 1 Report
Comments and Suggestions for Authors
Dear authors!
Most of my comments were carefully considered, and replies were relevant.
Best regards!
Reviewer 2 Report
Comments and Suggestions for Authors
Well done; thanks